# The impact of active and capable faults structural complexity on seismic hazard assessment for the design of linear infrastructures

Selina Bonini[1], Riccardo Asti[1], Giulio Viola[1], Giulia Tartaglia[2], Stefano Rodani[2], Gianluca Benedetti[2], Massimo Comedini[2], Gianluca Vignaroli[1]

[1]Department of Biological, Geological and Environmental Sciences, University of Bologna, Via Zamboni 67, 40126, Bologna, Italy

[2] ITALFERR S.p.A., Gruppo Ferrovie dello Stato Italiane – Architecture, Environment & Territory Department – Geology Division, Via Galati 87, 00155, Roma, Italy

*Correspondence to*: Selina Bonini (selina.bonini2@unibo.it)

**Abstract.** Since Active and Capable Faults (ACFs) may generate significant permanent deformation of the topographic surface, a careful evaluation of their spatial and geometric characteristics is essential for seismic hazard assessment when planning new linear infrastructures (e.g., roads, railway lines, pipelines). Although this is generally overlooked, the common structural complexity of fault zones leads to a non-uniform hazard along and across faults' traces, because of deformation partitioning. This study reviews the factors controlling fault rupture and propagation, specifically focusing on fault zone architecture and growth mechanisms. Four scenarios of physical interaction between ACFs and linear infrastructures are analysed. The fault-crossing scenario is likely the most susceptible to ground surface displacement, while the fault-parallel scenario needs evaluation of the width of fault damage zone overlapping with the infrastructure. Near-fault tip and transfer zone-crossing scenarios require specific assessment of the local deformation patterns. Given the importance of a structural geological approach toward the reliable assessment of seismic hazard related to ACFs, we review suitable investigations to derive appropriate deterministic geological constraints on the geometry, kinematics, slip and deformation style of ACF's. Our approach may have significant impact on the legislation regulating the early stages of infrastructural design.

## 1 Introduction

The accurate definition of site-specific parameters and processes related to active faulting is becoming of increasing importance to seismic risk assessment. Although ground shaking is universally considered as the primary cause of infrastructural damage, two large earthquakes occurred in 1964 (the Mw = 9.2 Great Alaska and the Mw = 7.4 Niigata earthquakes) have also highlighted the damaging potential coseismic permanent ground deformation (Youd 2014), including ground surface rupturing by slip along active and capable faults (ACFs). An ACF is defined by the International Atomic Energy Agency (2010) as a fault capable of producing significant displacement at or near the ground surface in response to the activation of a seismogenic source at depth and that has moved within the framework of the current tectonic stress regime (the Late Pleistocene is taken as the lower time limit for interplate faults). The importance of including and quantifying infrastructure safety in the context of earthquake-related hazard has become evident after the 1906 San Francisco earthquake (Mw = 7.9, Prentice and Ponti 1997), when the Wrights tunnel was damaged by a 1.8 m offset along the trace of a ruptured intersected seismogenic fault.

Fault rupturing is a complex phenomenon because of the common heterogeneity of the stress field acting along a fault zone, the variability of fault geometry and characteristics, the accumulated displacement history, and the lithological variability of the affected rocks (Ben-Zion and Sammis, 2003; Peacock et al., 2017; Treiman, 2010). In particular, the

geometric and displacement attributes of a fault zone that dynamically evolves through space and time are the direct expression of the structural mechanism(s) and the tectonic context that control the fault nucleation and progressive growth (Cartwright et al., 1995; Fossen and Rotevatn, 2016; Morley et al., 1990). If the fault is seismically active, these parameters also depend on earthquake magnitude as predicted by empirical scaling relationships (Ferrill et al., 2008; Kim and Sanderson, 2005; Leonard, 2010; Schultz et al., 2008; Vermilye and Scholz, 1998; Walsh et al., 2002; Wells and Coppersmith, 1994).

Until now, conventional approaches aimed at assessing seismic hazard in a given area have generally overlooked both the structural and geometrical complexities of active fault zones as well as the cross-cutting relationship between variably oriented faults and/or fault segments within bigger and more complex fault zones. Furthermore, distinct modes of deformation accommodation (i.e., rupturing along a single discrete slip surfaces or by diffuse off-fault deformation affecting a wider volume of rock) can lead to different patterns of permanent deformation at the ground surface, as evidenced by probabilistic fault displacement hazard analysis scenarios of ACFs (e.g. Moss and Ross, 2011; Petersen et al., 2011; Youngs et al., 2003).

The geohazard associated with the occurrence of ACFs in seismotectonically active areas is of particular interest to the design and construction of linear infrastructures (such as roads, railways, gas/oil pipelines, water adduction and distribution structures, and power lines) that, unlike punctual infrastructures (such as buildings, dams or nuclear installations), may extend over even hundreds of kilometres, thus potentially interacting with multiple tectonic features that possibly belong to very different tectonic contexts. Researchers engaging with this applied issue typically carry out seismic risk assessment primarily from an engineering and geotechnical perspective, thus dominantly focusing on the exposure or vulnerability of existing infrastructure networks (Shinoda et al., 2022; Zhu et al., 2020). There exists, instead, a remarkable knowledge and legislative gap regarding the hazard due to ACFs in the case of new linear infrastructures stretching in seismically active regions. An approach based on fault scaling laws has been recently proposed to estimate fault displacement hazard at lifelines-fault interference sites (e.g. Melissianos et al., 2023). That method, however, does not account for the complexity of fault characteristics and their spatial and temporal variability.

Numerous scientific studies have been and still are being conducted worldwide to learn more on ACFs. This constantly improving knowledge is being used to populate open-access databases that are fundamental to seismic hazard assessment (e.g. https://dggs.alaska.gov/pubs/id/24956; https://gbank.gsj.jp/activefault/index_e.html; https://data.gns.cri.nz/af/). The potential impact of seismic fault-related effects should be distinguished based on how ACFs geometrically interact and physically interfere with a linear infrastructure. Thus, a careful and quantitative analysis of the relationships between ACFs and linear infrastructures should be carried out already during early feasibility studies, as a support for the later infrastructure engineering design phase.

In this study, we analyse four possible geometrical interference situations between ACFs and linear infrastructures: fault-crossing, fault-parallel, near-fault tip, and transfer zone-crossing scenarios. In each case, we establish a) the main fault structural features to be defined and investigated, b) the related key fault parameters to be assessed and c) the most suitable investigations to derive appropriate deterministic geological constraints on geometry, kinematics, slip and deformation style of the ACF. By using as an example the national reference database for Italian ACFs (ITHACA Working Group, 2019; Figure 1) and relevant seismic microzonation legislations (Technical Commission on Seismic Microzonation, 2015), we propose an operative workflow to comprehensively parametrise an ACF. This study can inform ever more realistic scenarios of probabilistic fault displacement hazard analysis and, ultimately, aid in developing mitigation strategies to minimize infrastructure vulnerability.

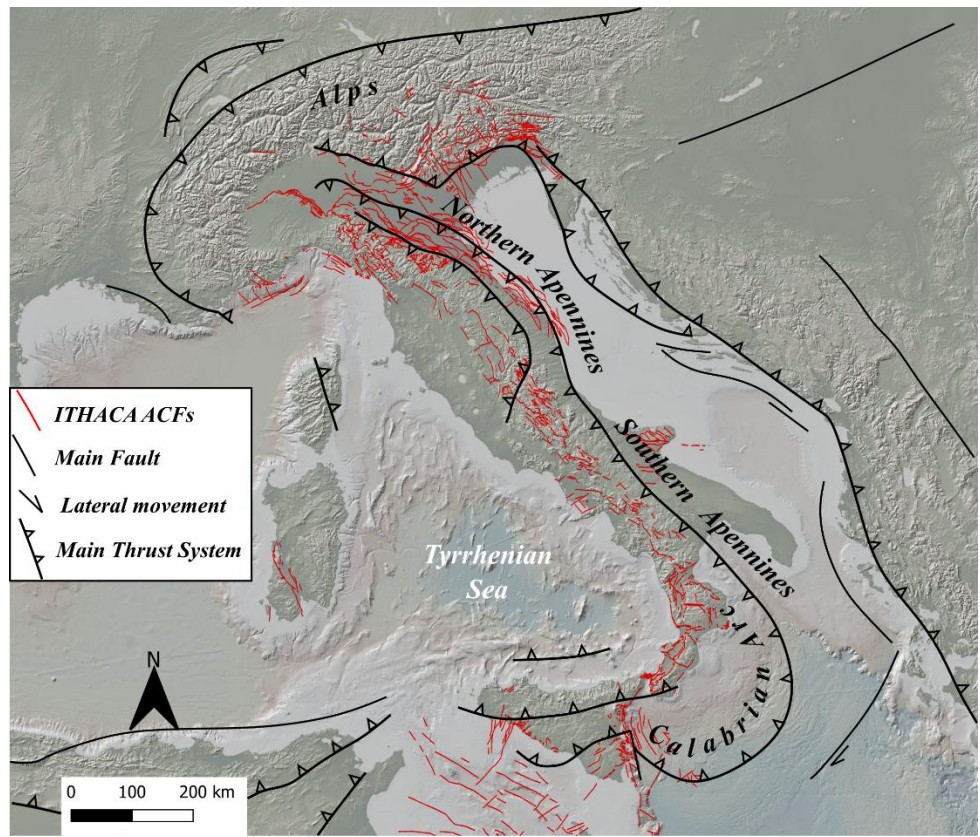

 **Figure 1: ACFs as per the ITHACA Italian catalogue** (ITHACA - ITaly HAzard from CApable faulting - A database of active capable faults of the Italian territory, 2024) **framed within the tectonic setting of Italy.**

## 2 Factors controlling fault rupturing and rupture propagation at the surface

Surface faulting is commonly associated with earthquakes with Mw ≥ 6, although the 2018 Lake Muir (Australia, Mw = 5.3, Clark et al., 2020) and 2019 Le Teil (France, Mw = 4.9, Ritz et al., 2020) moderate seismic events also ruptured the ground surface. Recently, two databases listing surface rupturing data from dozens of historical and instrumental earthquakes between 1872 and 2019 have been compiled from all over the world (SURE 2.0 and FDHI databases; Nurminen et al., 2022; Sarmiento et al., 2024). These databases clearly show that several factors steer the propagation of coseismic ruptures to the ground surface. In this regard, "external" and "internal" factors can be distinguished with respect to the fault system. External factors include the system characteristics that directly relate to the host rock and fluids. Rock rheology, for instance, is usually recognised as a controlling factor of rupture propagation, as narrow fault zones of high shear strain are typically associated with quartz-feldspathic rock types, whereas wide zones with diffuse high shear strain are commonly located in phyllosilicate-rich protoliths (Chester and Logan, 1986; Faulkner et al., 2003, 2008). Fault rupturing is facilitated during fluid-assisted deformation, as pore pressure reduces the normal stress on locked faults. Viscosity contrasts and the overburden thickness, in addition, control shear localization, which varies markedly passing from the bedrock to unconsolidated and/or water saturated sediments (Bray et al., 1994; Irvine and Hill, 1993; Johnson et al., 1997; Lazarte et al., 1994; Reid, 1910; Tchalenko, 1970).

Internal factors, on the contrary, represent all fault parameters that are connected with its kinematics, geometry, and mechanics. Fault orientation is known to exert significant impact on the kinematics of a given fault (Bott, 1959; Wallace, 1951), and numerous studies have shown that off-fault coseismic deformation is more likely localized in the hanging wall of inclined faults (Axen et al., 1999; Fletcher and Spelz, 2009; Huang and Johnson, 2010; Ma, 2009). Consequently, the kinematic characteristics of a fault affect the width of the surface rupture zone: a fault with pure strike-slip or pure dip-

slip movement is generally narrower than oblique slip faults. Other factors to consider are the coseismic slip magnitude and slip kinematics (Bray et al., 1994; Horsfield, 1977; Naylor et al., 1986; Quigley et al., 2012; Schlische et al., 2002; Tchalenko, 1970). In tectonically active areas, in fact, geometric parameters of a fault, such as coseismic length rupture and displacement recorded at the ground surface, are directly related to the moment magnitude of the earthquakes generated by it, as demonstrated by scale relationships (e.g. Wells and Coppersmith, 1994; Leonard, 2010). Figure 2 shows well-known databases of coseismic surface ruptures events integrated with data documented during historical and instrumental strong earthquakes (Table 1). Coseismic fault data (including maximum displacement and rupture length at the ground surface) exhibit a three-order-of-magnitude variation (from centimetres to tens of meters) for earthquake events ranging from magnitude 5 to 9 Figure 2a,b). This variation applies to all fault types (normal, reverse, and strike-slip) and can be attributed to factors influencing the propagation of fault rupture towards the topographic surface, along with the specific growth mechanisms governing each fault system (see below).

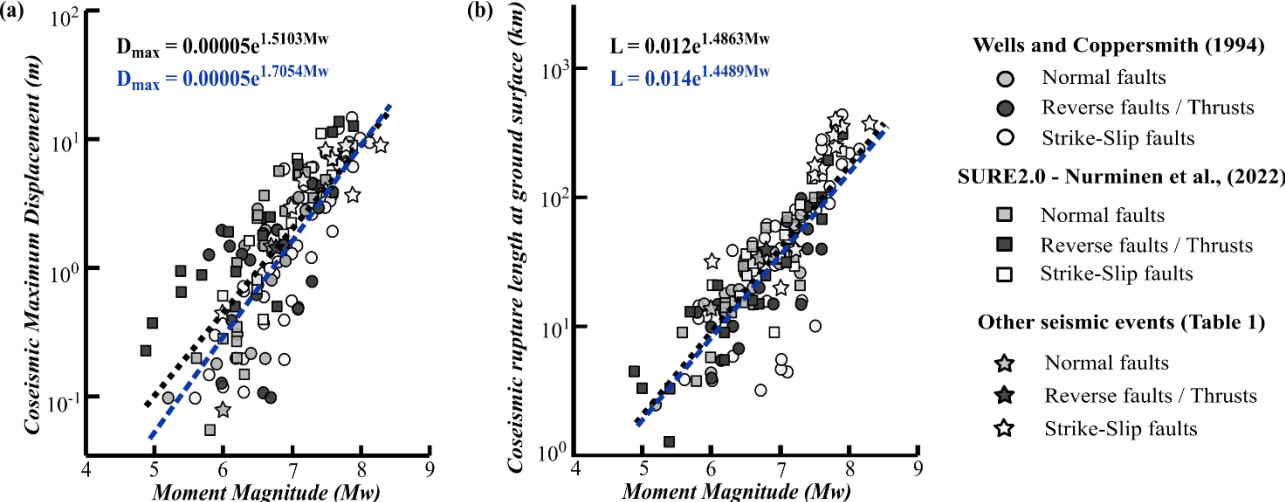

**Figure 2: Scaling laws correlating coseismic geometric fault attributes (maximum displacement and rupture length at the ground surface) and earthquake magnitude (a, b, respectively). Blue trend lines are calculated from** Wells and Coppersmith (1994)**, black trend lines are calculated considering also data from the SURE2.0 database** (Nurminen et al., 2022) **and Table 1.**

**Table 1: Historical and instrumental earthquakes that produced coseismic displacement at the ground surface and that are not included in** Wells and Coppersmith (1994) **and SURE2.0** (Nurminen et al., 2022) **datasets. L: length; $D_{max}$: maximum displacement.**

| Location | | Year | Mw | L (km) | $D_{max}$ (m) | Kinematics | References |
|---|---|---|---|---|---|---|---|
| **USA, CA** | Parkfield | 2004 | 6 | 32 | 0.46 | | Lienkaemper et al. (2006) |
| **Turkey** | Düzce | 1999 | 7.2 | 40 | 5 | | Wesnousky (2008) |
| **China** | Kokoxili | 2001 | 7.8 | 400 | 8 | | Lasserre et al. (2005) |
| **Mongolia** | Bolnai | 1905 | 8.3 | 375 | 9 | Strike-Slip | |
| **Turkey** | Erzincan | 1939 | 7.9 | 350 | 3.7 | | |
| **China** | Dongxi Co | 1930 | 7.5 | 150 | 4 | | Klinger et al. (2005) |
| | Manyi | 1997 | 7.6 | 170 | 7 | | |
| **Turkey** | Pazarcik | 2023 | 7.8 | 300 | 7.76 | | Dai et al. (2024) |
| | Elbistan | 2023 | 7.5 | 180 | 8.2 | | |
| **Armenia** | Spitak | 1988 | 6.8 | 40 | 1.6 | Reverse-Oblique | Philip et al. (1992) |
| **Italy** | Avezzano | 1915 | 6.7 | 35 | 1.2 | Normal | Galadini and Galli (1999) |
| | Colfiorito | 1997 | 6 | 14 | 0.08 | | Cello et al. (2000) |

## 3 Fault zone architecture and growth mechanisms

### 3.1 Fault zone architecture and attributes

Commonly, the first-order architecture of a fault zone (Figure 3) includes a central core and the enveloping damage zones (e.g. Caine et al., 1996; Cello et al., 2000; Chester et al., 1993; Chester and Logan, 1987). The fault core represents the product of highly localized deformation and most of the displacement within the faulted volume is accommodated therein (e.g. Bruhn et al., 1994; Childs et al., 1996; Sibson, 1977), as it is composed of multiple slip surfaces and fault rocks, such as fault gouge, breccia, and lenses of host rock (Torabi et al., 2019). Damage zones, instead, are characterised by relatively low deformation compared to fault cores. These zones generally exhibit several second-order structures such as subsidiary faults, fractures, veins, stylolites, cleavage, fault-related folds and/or drag folds (e.g. Berg and Skar, 2005; Billi et al., 2003; Bruhn et al., 1994; Faulkner et al., 2010; Odling et al., 2004).

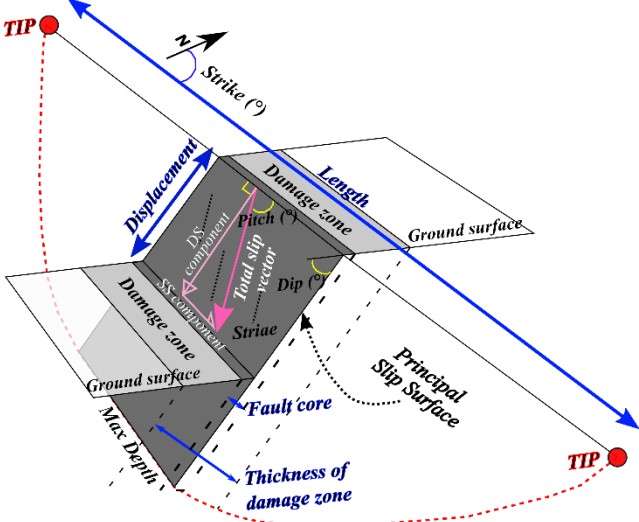

**Figure 3: Schematic illustration of a fault zone and definition of its main geometric and kinematic attributes.**

The thickness (or width, in map view) of these two structural domains, their strike and dip angles, length, maximum depth, and displacement represent the geometric attributes of a fault that need to be constrained when building the fault source model in seismic hazard analysis. However, the intricate nature of fault zones can lead to significant variability in fault parameters both along and across the fault zone (Figure 4), suggesting that variable mechanisms of nucleation and growth may contribute to fault zone evolution in space and time. This variability may thus have a substantial impact on the resolution and reliability of seismic hazard estimates.

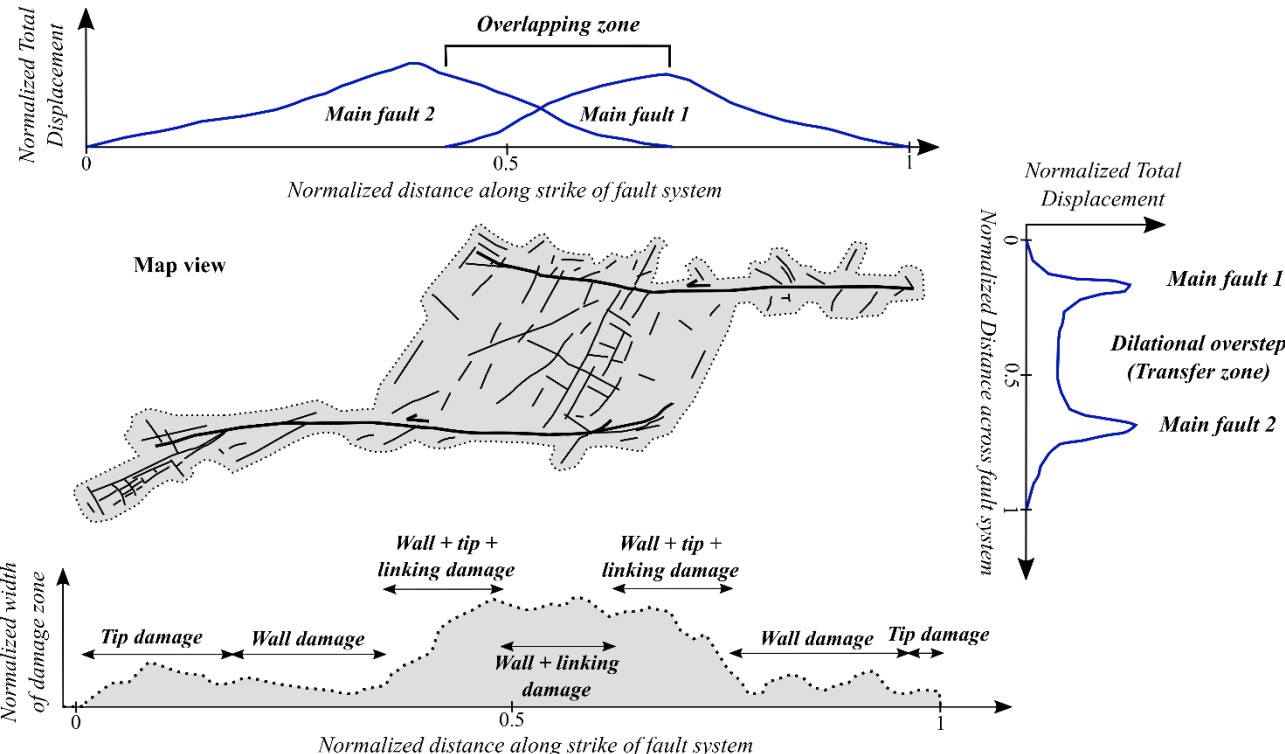

**Figure 4: Variable along- and across-strike displacement distribution (inspired by** Peacock and Sanderson, 1994**) and width of a complex fault zone. Terminology regarding damage zone types is from** Kim et al. (2004)**.**

Fault core geometry, for instance, varies in response to how deformation is localized in the rock volume, which is a function of the geomechanical properties of the host rock, the competency contrasts of the faulted lithotypes and of the presence of pre-existing anisotropies (e.g. Bastesen and Braathen, 2010; Childs et al., 2009; Foxford et al., 1998; Shipton et al., 2005, 2006; Sperrevik et al., 2002; van der Zee and Urai, 2005; Wibberley et al., 2008). In general, average values of fault core thickness vary between a few millimetres and one meter for an average (cumulative) displacement ranging from some centimetres to a few tens of meters (Johannessen, 2017; Torabi et al., 2019; Figure 5a). However, exceptional cases with fault displacements exceeding hundreds of kilometres are associated with cores up to ten to hundreds of meters thick (e.g. Childs et al., 2009; Wibberley et al., 2008).

Damage zones, on the other hand, include thicker rock volumes. Their width is usually defined by the frequency distribution of structures that commonly decreases with distance from the fault core (e.g. Chester and Logan, 1987; Goddard and Evans, 1995; Scholz, 1994; Smith et al., 1990). Many authors have related it to the cumulative displacement (e.g. Faulkner et al., 2011; Fossen et al., 2007; Torabi et al., 2020; Torabi and Berg, 2011; Figure 5b) or to the fault length (Vermilye and Scholz 1998 and references therein, in terms of process zone, Figure 5c). Scaling laws reveal that the width of a damage zone can vary from a few centimetres to some kilometres for cumulative displacements within a similar range. Scattered data suggest that the width of damage zones is typically two orders of magnitude smaller than the fault length.

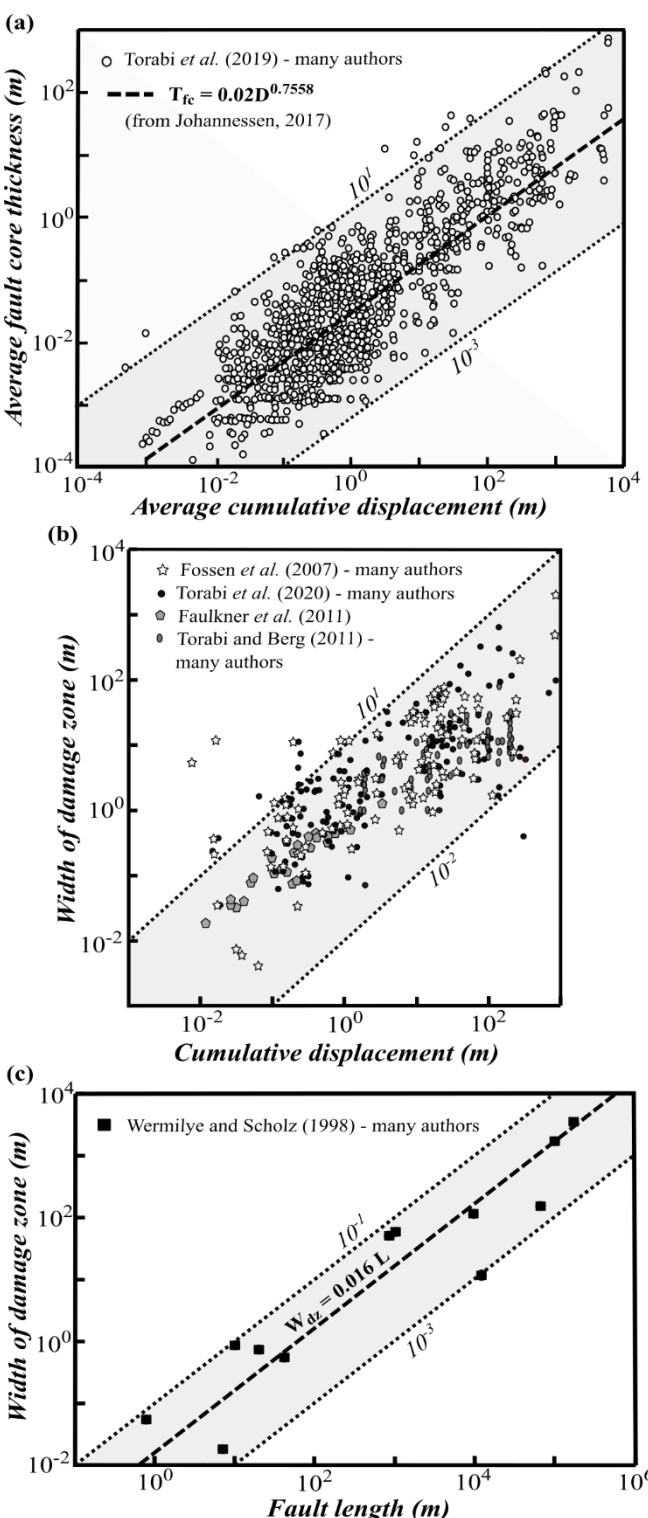

**Figure 5: (a) Fault core average thickness vs. average (cumulative) displacement** (power-law relationship from Johannessen 2017; data after Torabi et al. 2019 and references therein). **(b) Scaling law between fault damage zone width and cumulative displacement** (data after Fossen et al. 2007 and references therein; Faulkner et al. 2011; Torabi and Berg 2011 and references therein; Torabi et al. 2020 and references therein); **(c) Scaling law between cumulative offset and fault length** (data and power-law relationship after Vermilye and Scholz 1998 and references therein). **Dotted lines are traced from y = x.**

Damage zones are variably named depending on their location within and around the fault (*wall* vs *tip* damage zone), or between fault segments (*linking* damage zone) (Figure 4; Kim et al., 2004). A damage zone can become much wider

during fault interaction and may also include the tip damage zone, which usually contains structures that are misoriented to the main fault surface and that can accommodate shear movement or just tension opening (Kim et al., 2004, 2000, 2003; Kim and Sanderson, 2006; McGrath and Davison, 1995). For steeply dipping faults, subsidence and formation of fault propagation folds may cause the propagation of secondary structures preferably in the hanging wall (Berg and Skar, 2005; Evans, 1990; Ferrill et al., 2005), resulting in an uneven damage zone across the fault strike. Berg and Skar (2005) suggest a further subdivision of the fault damage zone in hanging wall and footwall damage zones, due to structural and geometric differences of secondary slip surfaces and fractures.

Fault displacement profiles, then, are controlled by several factors, including fault length (in map view), the fault aspect ratio (i.e., the ratio between fault length and height), fault shape (rectangular vs. elliptical), proximity of the fault to the free surface or other boundaries, configuration of far-field stresses, frictional and constitutive properties of the fault, variations in elastic properties and lithology along the fault, time-dependent rheology, near-tip processes, interaction with other faults, and fault segments linkage (e.g. Peacock, 2002; Schultz, 1999). Typically, displacement is considered zero at the fault tips and increases up to a maximum near the centre of the fault (Barnett et al., 1987). In the case of hard-linked fault segments, however, Peacock and Sanderson (1991, 1994) noted that the displacement gradient of a fault increases in the zone where fault segments overlap, and that the displacement maximum is no longer in the centre but located closer to the overlap zone (Figure 4).

Other fundamental fault attributes that have a relevant impact on seismic hazard assessments are fault kinematics (slip vectors and sense of shear), and state of activity (slip rate, recurrence interval for surface faulting, slip – total and per event – and triggering related earthquakes). Their estimation is necessary to fully parametrize a complex fault zone.

## 3.2 Fault nucleation and growth mechanisms

Following initial nucleation, the growth of a single fault can be ascribed to two main mechanisms (Figure 6): i) tip propagation and ii) displacement accumulation without significant tip propagation. Both growth processes refer to the development of a single, isolated fault.

In the case of the tip propagation model, faults form by developing a process zone, where micro-fractures form and coalesce along strike of the growing fault (Cowie and Shipton, 1998). Cox and Scholz (1988) demonstrated that Mode III shear cracks generate and link ahead of the crack tip. This leads to an increase in displacement proportional to the growth of fault length (e.g. "constant Dmax/L ratio model"), with maximum displacement values at the centre of the fault. Further fault evolution is accomplished by stress concentration at the tips, implying the development of various deformation features at the fault terminations, including wing cracks, horsetail fractures, fan-shaped branch faults, en échelon synthetic or antithetic shear fractures (Kim et al., 2004, 2000).

The second fault evolution mechanism leads to rapid growth and subsequent displacement accumulation without significant tip propagation ("Constant length model" by Walsh et al., 2003 or "Characteristic earthquake model" by Schwartz and Coppersmith, 1984). It is mainly applied to the development of a long fault structure above a buried reactivated fault (e.g. Giba et al., 2012). Slip in this case imposes an extension with deformation localizing in the cover above the fault, and overall fault propagation is upwards from the reactivated fault, generating tip bifurcation. It implies significant widening of the (wall-) damage zone.

When two subparallel, separated fault segments approach, they start to interact (e.g. Trudgill and Cartwright, 1994; Walsh et al., 2003). This is manifested by the curving of their terminations, bifurcation, development of a complex zone of subsidiary structures (splays, faults, fractures, deformation bands) or the formation of relay ramps (Fossen and Rotevatn, 2016). The interaction potentially evolves through time in linkage ("Fault linkage model" within fault population), causing

extensional or contractional oversteps (Kim et al., 2004). In this regard, Peacock et al. (2017) made a distinction between
i) approaching damage zones, if the interacting faults are kinematically linked but do not intersect, and ii) intersecting damage zones, which form around the intersection between two or more faults that abut, splay, or crosscut coevally (formed under the same tectonic regime) or sequentially (formed under different tectonic regimes). This second sub-category also includes the linking damage zone described by Kim et al. (2004) as regarding the area of deformation at a step between two sub-parallel coeval faults. When deformation is transferred from a fault to another, the intervening
deformation area is called transfer zone. Morley et al. (1990) introduced a systematic classification scheme for transfer zones, based on the dip direction (divergent or convergent) and the degree of overlap of faults between which the displacement transfer occurs. In transfer zones, deformation is distributed over a wider rock volume through diffuse faulting. The variable displacement transfer between shear structures and real fault zone depends on the overlap degree between two faults or fault segments. The interaction between faults is also related to the spacing and the total length of
the fault systems, depending on their kinematics. For example, laboratory experiments document that strike-slip faults interact when the spacing between two main faults is less than 10% of the combined fault length (An, 1997), while field data have been used to conclude that normal fault systems within the metre-to-kilometre scale range interact when the ratio between their minimum length and spacing is >14%  (Acocella et al., 2000).

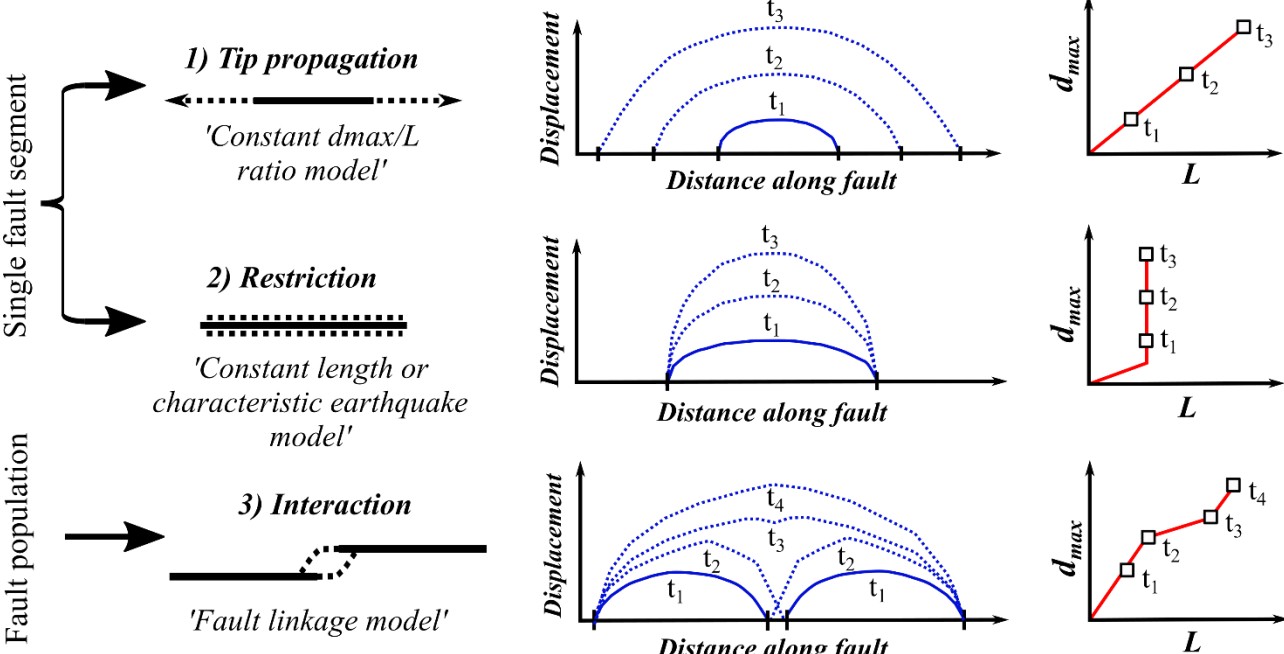

**Figure 6: Possible nucleation and growth mechanisms of faults. $t_1$-$t_4$ are sequential time increments during which a fault evolves.**

## 4 Geometric interference patterns between ACFs and linear infrastructures

In this section, we apply the fault attributes and growth mechanisms described above to an ideal scenario of ACF evolution. Firstly, we analyse the impact of these mechanisms upon the four most likely geometric interference patterns between an ACF and a linear infrastructure. Then, we define a spectrum of structural geological parameters that should
be carefully monitored at the interference zones (IZ) between an ACF and the linear infrastructure (Table 2). We define the IZ as the mappable area where the deformation effects associated with ACFs activity are expected to impact the infrastructure.

We have considered the following fault-linear infrastructure interference patterns (Figure 7): i) *Fault-crossing infrastructure*, if the linear infrastructure intersects the trace of a main fault at any incidence angle along its length; ii) *Fault-parallel infrastructure*, if the linear infrastructure runs essentially parallel to the main fault trace, potentially interfering with the ACF damage zone and/or its core; iii) *Near-fault tip infrastructure*, if the linear infrastructure runs at or near the influence lobes characterizing the ACF tip points; iv) *Transfer zone-crossing infrastructure*, if the linear infrastructure passes by an area located between two or more interacting ACFs or ACF segments.

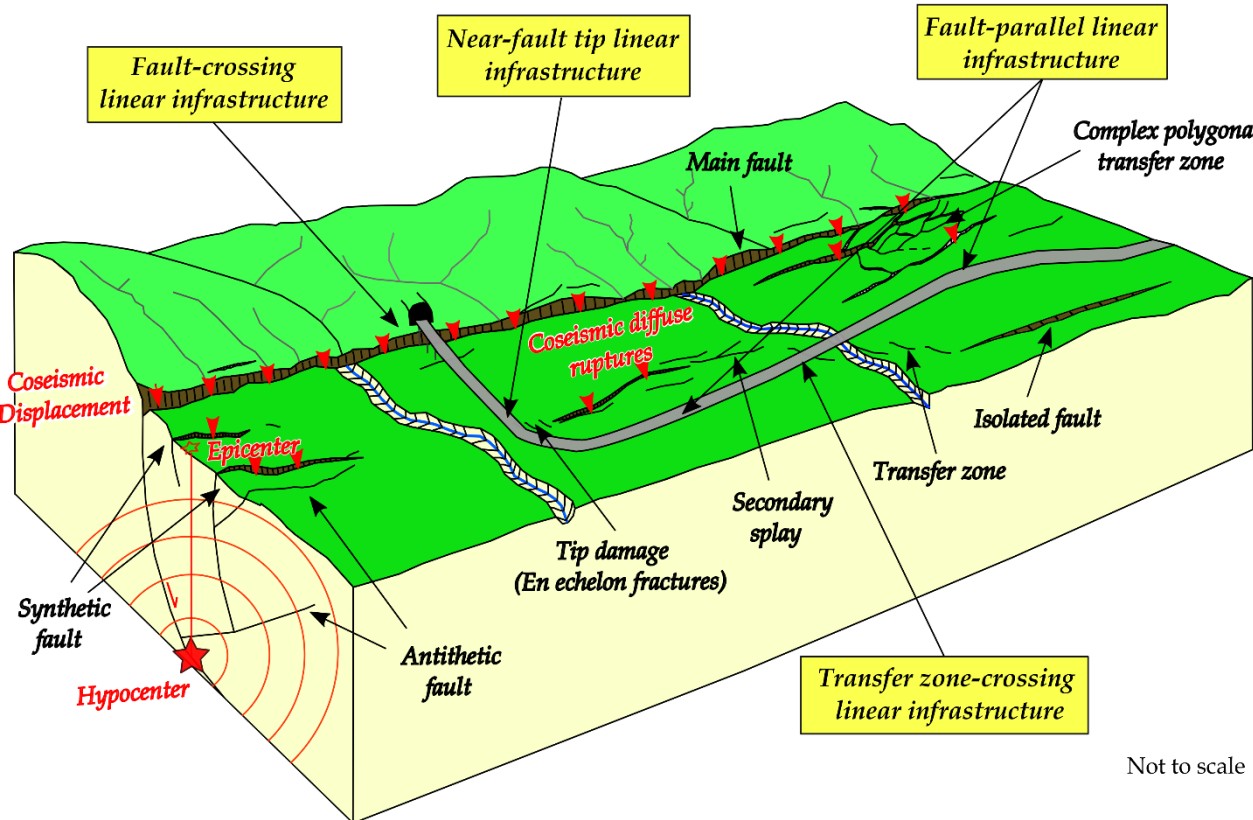

**Figure 7: Schematic diagram illustrating the possible geometrical interference patterns between an ACF and a linear infrastructure.**

Since the crossing of different portions of an ACF by a linear infrastructure implies that some fault parameters may weigh more than others, each scenario requires a distinct assessment of the fault parameters associated with the corresponding ACF element. Thus, we can prioritize some fault parameters over others in fault hazard assessment (Figure 8).

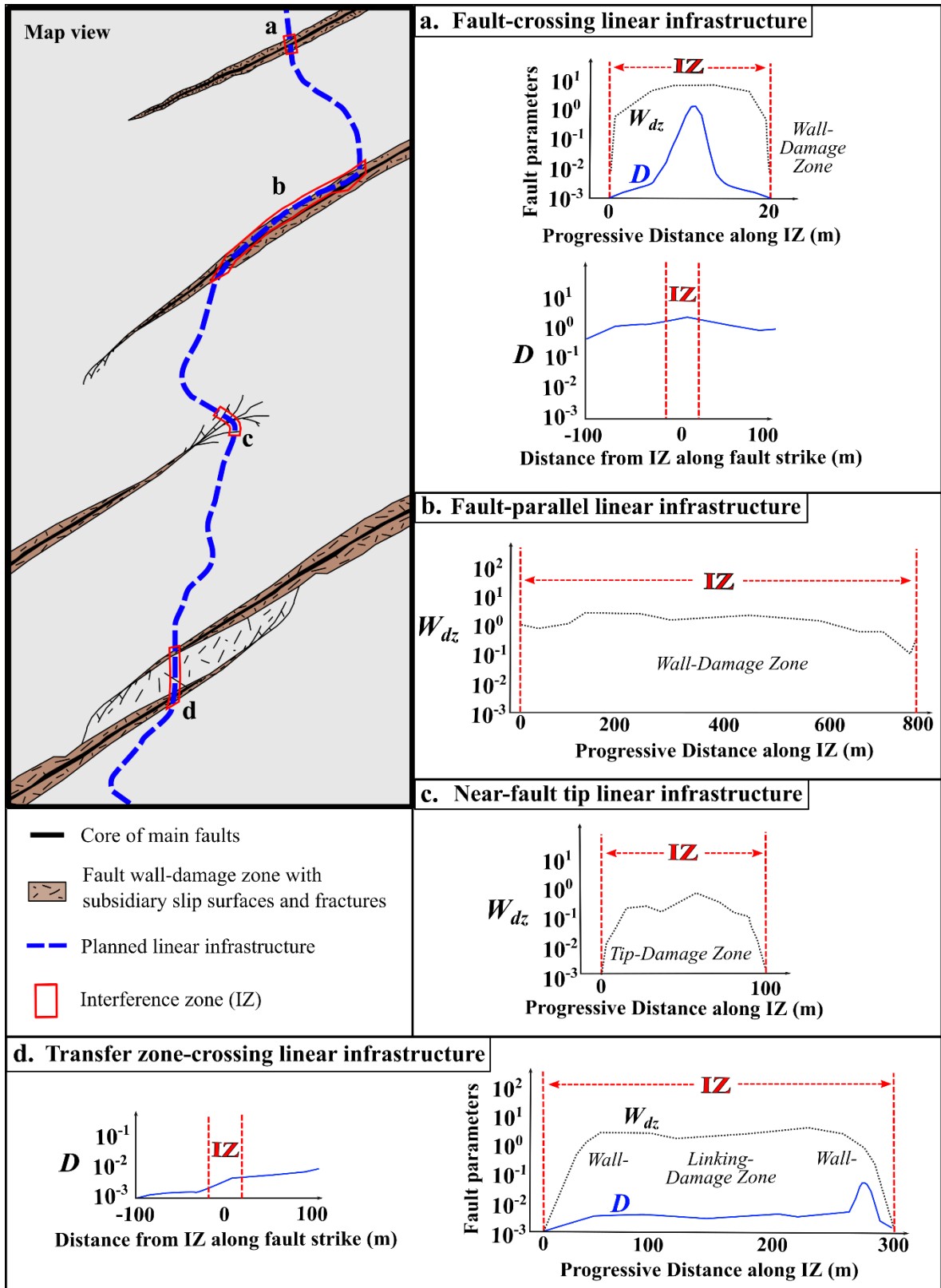

**Figure 8: Examples of geometric parameters that can support the definition of the four fault–linear infrastructure interference scenarios discussed in the text (D = displacement, Wdz = width of damage zone, in graphs a-d). Drawing not to scale.**

In a *fault-crossing* scenario it is particularly important to investigate (i) the coseismic displacement at the ground surface, especially if the offset is accommodated on thin slip surfaces, and (ii) the deformation accommodated within the damage zone. (i) While in fault source models a unique mean value of slip is usually considered for the entire ACF because of the


simplified nature of such models, displacement can be variable along the strike of a fault (e.g. Peacock, 2002; Schultz, 1999); Figure 4). Both fault growth mechanisms (single fault segment and fault population; Figure 6) predict that the maximum displacement occurs at the centre of the fault system, while it gradually decreases to zero at the fault tips. (ii)

Based on scaling relationships (Figure 5b), the width of the damage zone scales to the cumulative displacement through a power-law relationship. Thus, the maximum width of the damage zone is expected to coincide with the zone of maximum displacement at the centre of the fault system. From the consideration above, it is reasonable to expect that the potential impact linked to coseismic displacement and damage zone development can be higher if the linear infrastructure crosses the central zone of a fault system (either fault segment or interacting fault segments), rather than near the tips

(e.g., Barnhart et al., 2015; Rockwell et al., 2002). To obtain high-resolution data on both amount of displacement and damage zone width at the IZ, it is crucial to analyse the distribution of deformation both along fault strike (for displacement) and across the fault strike (for damage zones; Figure 8a).

The extent of the study area on both sides of the linear infrastructure should be calculated considering the scaling relationships in Figure 2a,b, and Figure 5c. During design stage, the area usually included in the detailed study extends

for few hundreds of meters away from the linear infrastructure, on both sides. For example, this extent is probably more than sufficient in the case of the Italian territory. In fact, coseismic deformation is expected to affect an area of maximum tens of meters, with the occurrence Mw = 7 earthquakes, which is, at the moment, the highest magnitude value reported in the historical seismological record of Italy. However, this distance from the infrastructure might be inadequate in the case of stronger earthquakes.

In a *fault-parallel* scenario, it is critical to verify if the infrastructure interacts with the fault damage zone (Figure 8b) and, eventually, to evaluate the length of the infrastructure section falling within the IZ. Similar to the previous scenario, constraining the width of the damage zone requires considering the position of the IZ relative to the length of the main fault surface (central part or toward the tips). Scaling laws suggest that the width of the damage zone is c. a couple of orders of magnitude lower than fault length (Figure 5c) and thicker in the hanging wall of dipping faults.

In a *near-fault tip* scenario, the deformation associated with fault tips is particularly significant. Deformation structures at the IZ stem from how the fault accommodated displacement at its tips as it progressively grew in length over time (fault growth mechanism by tip propagation in Figure 6). It is, hence, a priority to evaluate the extent of the area that is affected by fractures (tip damage zone), including its potential further growth during future earthquakes (Figure 8c). For high angle and steeply dipping faults, the assessment of deformation near the fault tips becomes even more critical, as the tips

of these faults can be associated with a broad range of fracture patterns, depending on the fracturing propagation mode and the deviatoric stress existing at the tips (Kim et al., 2004, 2000, 2003; Kim and Sanderson, 2006; McGrath and Davison, 1995).

The *transfer zone-crossing* scenario implies the interaction between two (or more) active faults where deformation is accommodated both along each single fault segment as well as in the fault overstep region. This complex structural setting

encompasses all fault mechanisms and variations of fault characteristics described previously. In this scenario, the complexity of the IZ is linked to the width of the damage zone that develops during fault interaction (Figure 4). It is necessary to consider both the extent of the zone affected by diffuse deformation within the damage zone (wall damage zone plus linking damage zone) and the fact that there are discrete surfaces potentially accommodating most of the displacement. The priority should be the quantification of both the width of the transfer zone-related damage zone crossed

by the linear infrastructure and the slip accommodated by the main fault surfaces (if crossed; Figure 8d). During planning, in fact, it is essential to recognize whether the estimated coseismic displacement could be accommodated by fault zones a few meters thick, rather than by secondary planes within an extensive damage zone hundreds of meters thick. Therefore,

the same considerations made for the assessment of the displacement are applied. Transfer zones can moreover exhibit complex arrays of synthetic and antithetic shear fractures, which can define blocks that can potentially rotate during coseismic rupturing (Kim et al., 2000). If these blocks have a considerable dimension with respect to the linear infrastructure (e.g., decametric scale), the probability of discrete rotation during ACF activation should also be included in the fault hazard assessment.

From a geometrical standpoint, the angle of incidence between the linear infrastructure and the ACF represents a first-order parameter in defining the extent of the IZ. As a preliminary assessment, Figure 9 qualitatively illustrates the relationships between the incidence angle and the IZ for different interference scenarios. Specifically, lower incidence angles (i.e., 10 – 45°) are associated with broader IZs. In contrast, a near-perpendicular intersection typically limits the extent of IZ. However, if the linear infrastructure runs subparallel to the ACF, thus running within the fault damage zone, the entire infrastructure segment may fall within the IZ.

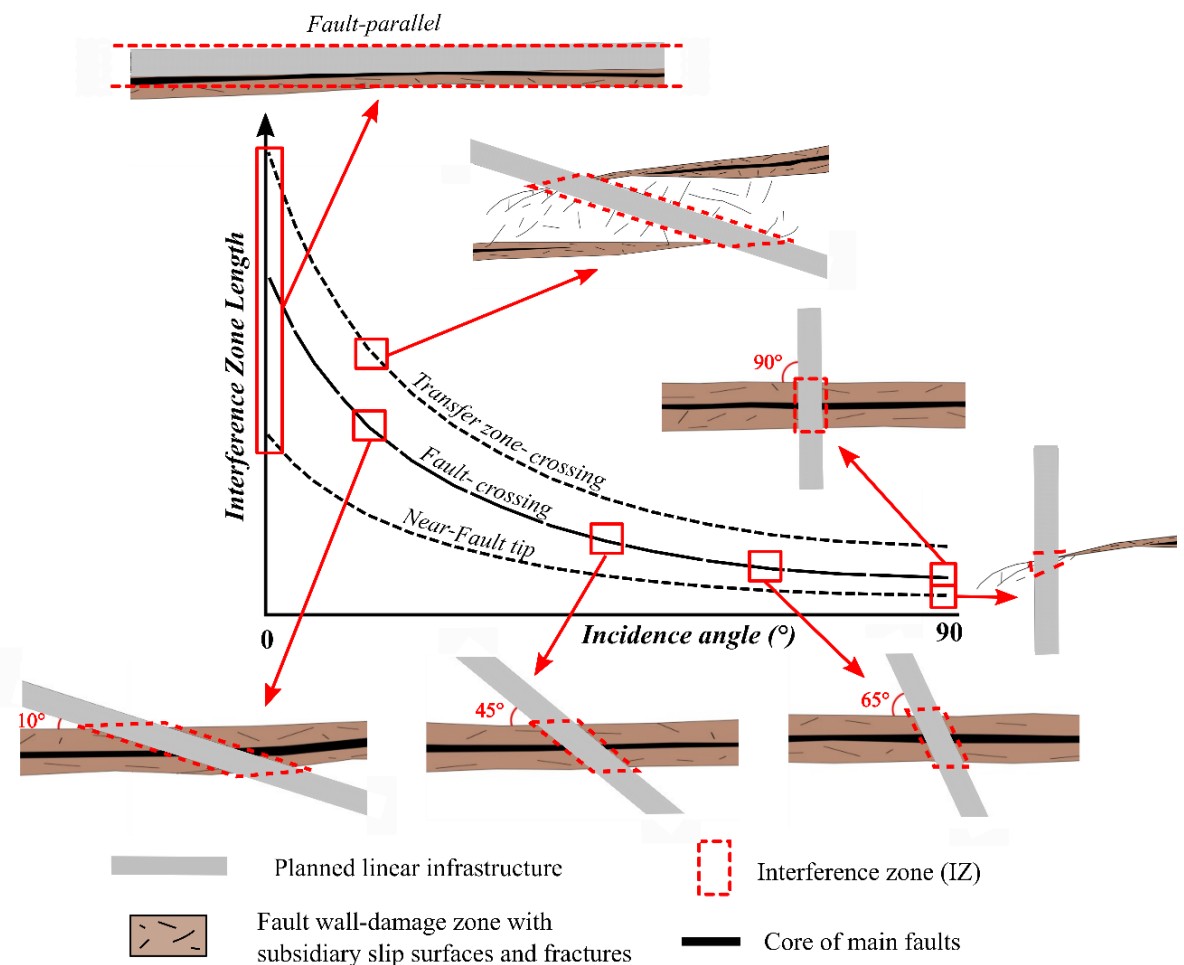

Figure 9 - **Variation of the area potentially affected by ACF effects (dotted red area) as a function of the incidence angle and the ACF-linear infrastructure scenario.**

## 5 Discussion

### 5.1 ACF parameters characterization

To date, neither scientific studies nor international guidelines (e.g. International Atomic Energy Agency, 2010) have proposed a procedure to assess and characterize the structural complexity of ACFs as a support to seismic hazard assessment in infrastructural design. In Italy, existing guidelines for seismic microzonation studies (Technical

Commission on Seismic Microzonation, 2015) prescribe the mapping of ACFs with the aim to outline respect areas in map view linked to the kinematics of the fault (ratio footwall/hanging wall = 1:4 for normal faults; 1:2 for reverse faults). However, nothing specific is suggested/recommended concerning the characterization of ACFs and the variation of their parameters in space.

Since ACF behaviour and attributes are strictly connected to the mechanisms by which the fault has developed, we propose an approach to optimize hazard assessment according to fault structural complexities and fault-linear infrastructure interference patterns. The assessment of seismic hazard due to the occurrence of ACF(s) should, therefore, address the study of specific fault attributes for each of the considered interference scenarios. This can be useful in a very preliminary phase of the design stage such as the feasibility study, especially for linear infrastructures of considerable length intersecting several potential ACFs.

Our study shows that:

- Coseismic displacement and damage zone width are crucial fault parameters, that are strictly linked to the structural architecture of an ACF. Their variability is related to the specific growth mechanism governing the development of the ACF (Figure 6), which we consider as first-order deformation processes driving the spatial-time evolution of fault zones. The study of these growth mechanisms should be combined with that of the factors controlling rupture dynamics and the surface ruptures pattern (that is sustained by the structural setting), as well as site-specific conditions (e.g., water content, topographic slope, or soil/rock cohesivity), to enhance our understanding of the spatial distribution of fault attributes within the IZ.

- How an ACF geometrically interacts and physically interferes with a linear infrastructure defines distinct potential types of influence on the seismic fault-related effects. This emphasizes the importance of considering these pattern interactions and prioritizing the characterization of key fault aspects for the four scenarios presented above. Figure 10 proposes a summary of the main ACF parameters and their respective priority level for each scenario.

- The parameterization of the geometric properties of the fault zone in Figure 5a,b,c provides a general, first approximation of the area potentially affected by coseismic rupture for a given earthquake magnitude, including the development of distributed ruptures. However, due to the significant uncertainties of the given empirical relationships, these estimates should be integrated by site-specific investigations in order to refine the fault model and, consequently, to improve the definition of the spatial extent of the IZ.

- If the full parametrization of ACFs cannot be performed due to either inaccessibility to the site or low quality of the outcrops, we should rely on fault scaling laws (Figure 2a,b) to predict the expected range of coseismic displacement along an ACF for a given maximum earthquake magnitude (e.g. Leonard, 2010; Wells and Coppersmith, 1994). The outcome could offer an indication of the magnitude of the expected coseismic offset (average and maximum) at the ground surface that could affect the infrastructure.

- The selection of fault attributes relevant for the parametrization of the fault-linear infrastructure interference patterns is not affected by the kinematic class of the ACF.

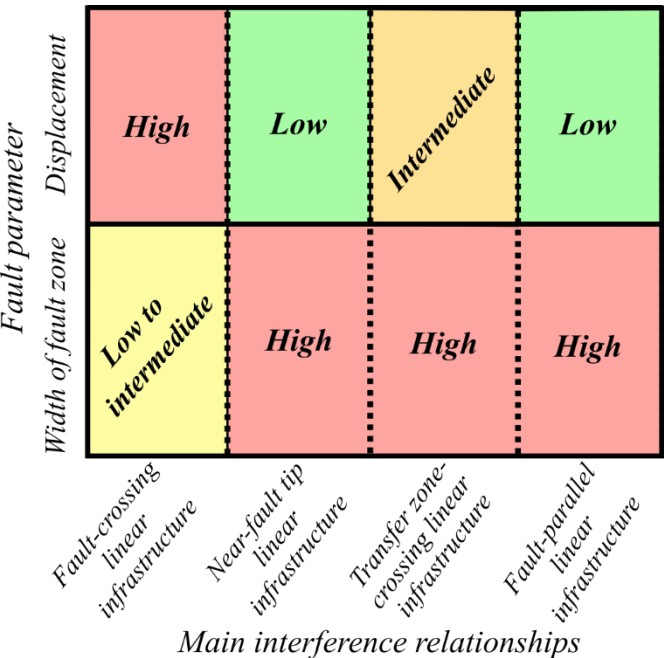

**Figure 10: Schematic correlation between fault parameters and fault-linear infrastructure interference pattern. For each scenario, the recommended priority for assessing specific ACF parameters is qualitatively indicated.**

**5.2 Toward a structural-based approach for the Seismic Hazard Assessment in Linear Infrastructure design**

In order to characterize the style and mechanisms governing the structural complexity of an ACF at fault-linear infrastructure interference zones, we propose a structural geological approach that is to be implemented during the design phase of an infrastructure project (Table 2). This approach supports a multidisciplinary and multiscalar work that integrates traditional methods (e.g. aerophotogrammetry, paleoseismology) and new developments (e.g. Remote Sensing analysis, geophysical investigations) for the characterization of areas affected by active tectonics (McCalpin, 2009, 2013),

together with geological and geotechnical investigations. These methods may estimate slip rates over different time scales (Cowie and Roberts, 2001) such that we should carefully consider which is the best method to be used for a specific scenario of seismic hazard assessment. It is noteworthy that a geological and structural study of the site of interest is a fundamental step, that permits to collect quantitative structural data and ground-truth all the data collected by indirect techniques.

Once the presence of a potential ACF has been ascertained (for example, by relying on dataset of ITHACA and/or seismological databases, seismic microzonation studies, fieldwork evidence, geological cartography, and/or scientific literature), detailed investigations should be started at a smaller scale, aiming to evaluate the geometric ACF-linear infrastructure interference scenario and what portion of the ACF can potentially involve the planned linear infrastructure. This evaluation represents the starting point of a structural geology-based seismic hazard assessment. The suggested

operative workflow can be enriched with a morpho-tectonic study performed with remote sensing data. Remote sensing is a useful indirect method to perform morpho-structural analyses over large areas. It can rely on the analysis of digital terrain models (DTMs) or LiDAR (Light-Detection and Ranging; laser scanning) data, the latter being capable of reproducing DTMs by filtering the vegetation cover. High resolution LiDAR images can be used to evaluate fault zone width, by identifying off-fault deformation features, rupture complexity, and post-seismic phenomena (e.g. Arrowsmith

and Zielke, 2009; Cunningham et al., 2006; DeLong et al., 2010; Nissen et al., 2012). These tools are particularly suitable for *transfer zone-crossing* scenarios, which require the extent of the area affected by fault deformation to be determined.

Remote sensing can be used as a starting point for *fault-parallel* and *near-fault tip* scenarios. Unfortunately, the method is limited by the resolution of the data - usually LiDAR with a spatial resolution of 1 m being the best available (e.g. Brunori et al., 2013). As a result, the assessment of both damage zone width and displacement can be underestimated, especially for structures in the decimetre-to-centimetre range. It is important to note that this approach only works if the topographic expression of fault-related deformation is preserved. In areas where erosion or anthropogenic activities have obliterated or concealed the fault signal, remote sensing may underestimate both the width of the damage zone and the amount of displacement. Remote sensing can be convenient for remote areas and/or in a very initial phase of the study. Nevertheless, such results should be compared with the information reported on official geological cartography and subsequently verified through a fieldwork study, as ground-truthing is essential for detecting and characterizing centimetre-scale deformations.

Remote sensing can also be used to estimate the cumulative offset by measuring the height of fault scarps (e.g. Haddad et al., 2012) and horizontal displacement of geomorphic features (e.g. Bihong and Yasuo, 2007); it can therefore be also useful to perform preliminary hazard assessment for the *fault-crossing* scenarios. However, considerations derived from remote sensing must be preferably integrated by field observations. Geomorphic methods can be used to estimate throw and slip along faults by studying appropriate landforms that can be directly measured in the field and potentially expanded with aerophotogrammetry and terrestrial LiDAR (e.g. Wilkinson et al., 2015). When such landforms can be associated with reliable time constraints (e.g. radiometric dating of associated sediments), values can be converted in throw and slip rates (e.g. Nissen et al., 2009).

Once the fault trace has been recognized, a structurally informed geological survey allows to strengthen the study with surface data and to constrain the fault geometry and kinematics with deterministic geological parameters (e.g., fault plane orientation, dip angle, pitch, displacement from cross correlation of depositional units, etc.), especially in case when the fault reaches the topographic surface. In such a case, a detailed evaluation of the displacement distribution along strike of the main fault surface may be possible.

If the fault is buried under a few meters of recent sediments, on the other hand, it is necessary to find the exact location of the main fault surfaces (e.g., Liu et al., 2008). In active tectonics studies, geophysical methods offer a useful tool in areas where it may be difficult or even impossible to map/study the surface expression(s) of active faults (for instance in the *near-fault tip* scenario, where faulting may be accommodated by multiple smaller splays that rupture simultaneously during earthquakes). These methods are also essential when information about the subsurface continuation, structure, or characteristics of faults is required. In all cases, correlations with field data are crucial to validate and interpret the observed geophysical signals. Geophysical surveys, when combined with geological and geotechnical investigations, not only help to delineate the subsurface geometry of the fault but also facilitate the identification of subtle deformation features that might otherwise be missed (e.g., Giocoli et al., 2008). Ground Penetrating Radar (GPR) and Electric Resistivity Tomography (ERT) are also functional to establish the exact point where paleoseismological trenches should be executed (e.g., Galli et al., 2006; Liner and Liner, 1997; Salvi et al., 2003; Storz et al., 2000; Suzuki et al., 2000; Wyatt et al., 1996).

The most direct techniques for evaluating the displacement recorded on fault surfaces are paleoseismological studies through trenching and radiometric dating (Galli et al., 2008; McCalpin, 2009; Pantosti et al., 1993). Excavating trenches at specific points across faults, especially in relatively recent sediments, allows for a detailed examination of the stratigraphy of syn-tectonic units, fault-related structures, and displaced geological layers. This can lead to good estimates of the amount of past earthquake-induced fault displacement, recurrence time, slip rates, and magnitude. It should be noted that this would be a punctual value, probably related to the internal and external factors controlling fault propagation

up to the ground surface discussed in Sect. 2. For this reason, results may not be representative for the entire ACF but only for the specific analysed fault segment (e.g., Iezzi et al., 2023).

At this point, finally, to improve our understanding of faulting within an absolute time frame, specific geochronological constraints could be derived (Hocking et al., 2017; Schimmelpfennig et al., 2009). This can be achieved by the U-Th direct dating of fault movement recorded, for example, by syn-tectonic mineralizations (e.g. Uysal et al., 2009; Vignaroli et al., 2022) or exposure of active fault scarps using cosmogenic nuclides dating like $^{10}$Be, $^{36}$Cl, $^{10}$He (e.g. Benavente et al., 2017; Mozafari et al., 2019), or rare earth element analyses (e.g., Bello et al., 2023). Moreover, Optically Stimulated

Luminescence applied to quartz (OSL) can be used for dating sediments (e.g. Porat et al., 1997, 2009; Rockwell et al., 2009) and fault gouge (e.g. Tsakalos et al., 2020), usually combined with radiocarbon dating (e.g. Ferrater et al., 2016; Vargas et al., 2014). By systematically applying these techniques, we can potentially derive constraints on fault location and geometry, average time of recurrence and slip rate (mm/yr) of ACFs, to better evaluate the hazard linked to ground surface displacement.

**Table 2: Main interference relationships between ACF and the trace of a linear infrastructure. Direct and indirect methods suitable for their investigation are proposed.**

| Interference relationships | Crossing | | | Not crossing |
|---|---|---|---|---|
| | *ACF-crossing linear infrastructure* | *Transfer zone-crossing linear infrastructure* | *Near-fault tip linear infrastructure* | *ACF-parallel linear infrastructure* |
| *Main fault aspects to be detailed* | Fault displacement recorded on both principal slip surface in the core and subsidiary faults, cumulative offset. | Width of interacting-damage zone, displacement associated with transfer faults if necessary | Width of tip-damage zone | Width of the damage zone. If the linear infrastructure is located out of the damage zone, it is not necessary to proceed with a detailed characterization |
| *Investigations* | Structural survey in the field, aerophotogrammetry, geophysical investigations (GPR, ERT), paleoseismological study | Remote Sensing analysis (DTMs, orthophotos, high resolution LiDAR images), aerophotogrammetry, structural survey in the field, paleoseismological study | Remote Sensing analysis (DTMs, high resolution LiDAR images), aerophotogrammetry, structural survey in the field, geophysical investigations (GPR, ERT) | Remote Sensing analysis (DTMs, orthophotos, high resolution LiDAR images), aerophotogrammetry, structural survey in the field |

## 6 Conclusions

ACFs characterization is becoming of increasing importance to seismic hazard assessment during the planning of linear infrastructures, as they can produce permanent deformation of the ground surface. Thus, the feasibility phase of a new

linear infrastructure should carefully consider the structural study of these complex tectonic structures, which are very common in tectonically active regions, such as most of the Italian territory. This impact can result from a number of factors related to the structural complexity of the internal architecture of the ACFs, which includes fault core, wall-damage zone, tip-damage zone and transfer zones between two or more faults (or fault segments). As these aspects vary significantly along and across strike of a fault system, it follows that fault-related hazard can vary and, in certain cases,

be related more to a specific parameter than others. Consequently, the hazard value also depends on the portion of the fault zone being crossed by the planned linear infrastructure. We distinguished four possible interference relations between

the planned infrastructure line and an ACF. For each case, we determined: a) the main fault structural aspects to be detected with higher priority; b) which parameters weigh more in the hazard evaluation; c) the most suitable investigations to employ, considering that the geological and structural study in the field remains a fundamental step. This is expected to provide a new structural geology-based methodological approach for the hazard assessment related to ACFs, which can support infrastructural planning in the preliminary stages of a feasibility project study. This approach may be more compatible with the timing and budget requirements associated with design projects, as some dating, aerial acquisition, data processing and interpretation could be carried out within shorter timeframes than those allocated in infrastructure construction schedules. At the same time, it appears to be more detailed and complete than conventional seismic hazard assessment methods, since it considers with increased resolution the structural complexity of fault zones. Furthermore, predicting a certain level of seismic hazard along the linear infrastructure represents a key step in planning, which contributes to improved resilience of the infrastructure itself.

**Author contribution**

SB: Conceptualization, Investigation, Writing – original draft preparation, reviewing & editing, Validation, Visualization, Methodology. RA: Writing – reviewing & editing, investigation, Supervision. GV: Supervision, Writing – review & editing, Validation, Funding acquisition. GT: Writing – reviewing & editing, Validation, Supervision. SR and GB: Validation, Supervision. MC: Validation, Supervision, Funding acquisition. GV: Conceptualization, Project administration, Validation, Supervision, Writing – review & editing, Funding acquisition.

**Competing interests**

The authors declare that they have no conflict of interest.

**ACKNOWLEDGEMENTS**

The manuscript benefits from the comments and suggestions provided by Dr. Robin Kurtz and the anonymous Reviewer. We thank the Editor Dr. Veronica Pazzi to handle the manuscript. Dr. Oona Scotti is acknowledged for providing us useful advice. This work belongs to the PhD project of S. Bonini "Parameters for assessing active faulting and related surface effects in railway design". The project is funded by a research agreement between ITALFERR S.p.A. and BiGeA Dept (CONTRATTO PER IL COFINANZIAMENTO DI BORSE DI DOTTORATO ATTIVATE AI SENSI DEL DM 352 DEL 9 APRILE 2022 - 38° CICLO - A.A. 2022/2023). This work is also partially supported by the PE3 RETURN Project (CUP J33C22002840002; R.A. and G. Vignaroli).

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
