# Peer review of "The impact of active and capable faults structural complexity on seismic hazard assessment for the design of linear infrastructures"

_EGUsphere, 2025_

## Author Response (AR1)

**RESPONSE TO THE REVIEWERS' COMMENTS**

**REV#1 (anonymous)**

**Reply to general comments**

*COMMENT: This is a useful and well-written paper about the impact of active and capable faults structural complexity on seismic hazard assessment for the design of linear infrastructures. However, a moderate improvement is needed in the structure of the paper before its acceptance for publication. Some contents of sections 4 and 5 are more results than methodological explanations or discussion. Therefore, I recommend a new section, "5. Results," needs to be created where the authors may present their results, taking relevant material mainly from sections 4 and 5.*

REPLY: We thank the Reviewer for the valuable comment and suggestions. We fully acknowledge the importance of clearly highlighting the original and innovative contribution of our work within the framework of seismic hazard assessment for infrastructure planning purposes. In this context, we would like to clarify that the main aim of the present study is to introduce and discuss the implications of the structural complexity of active and capable faults (ACFs) on seismic hazard evaluations, specifically in relation to the design of linear infrastructures. Our primary objectives are: (i) to propose conceptual scenarios illustrating the potential physical interactions between ACFs and linear infrastructures; and (ii) to analyse the key structural factors that govern the deformation patterns interfering with such infrastructures.

It is important to note that this work does not introduce new geological, structural, or geophysical data. Our primary focus is, instead, to conceptually analyse and discuss the geometrical configurations that may arise from ACF-linear infrastructure interactions (Section 4). From this analysis, we elaborate and then share with readership a practical workflow to deal with the comprehensive parametrisation of an ACF (Section 5). We carefully assessed the request by the Reviewer and seriously considered alternative structures to the paper but, in the end, convinced ourselves that the current organisation of the manuscript effectively reflects our methodological approach and the conceptual nature of this contribution. Therefore, we hope that it is accepted that we prefer to maintain the current text organisation in the revised version.

However, in order to improve the flow of the part of the discussion dealing with the size of the Interference Zone (IZ), we have decided to move former Figure 10 (now Figure 9) to Section 4 (Lines 298-303). We believe this change provides a more coherent sequence of information and better supports the reader's understanding of the proposed conceptual framework.

**REV#2 (anonymous)**

**Reply to general comments**

*COMMENT 1: The definition of the IZ represents a key point of the proposed process of hazard assessment for linear structures, hence a proper scheme showing the IC dimention compared to the fault zone might improve the consistency of the concept. Its definition is suggested to be based on scaling law, which is a very good idea for a first order assessment. The issue is that those scaling laws present several orders of magnitude, and the definition of the considered dimension remains very coarse. I am wondering about the uncertainties related to the definition of those lengths, and how to i. constraint them and ii. Give some security range (kind if geometric safety factor), to correctly constrain the dimension of the zone, minimizing in the one hand the probability of missing some fault surface rupture and associated offsets, and in the other hand do not exaggerate the IZ size, as the cost possibly involved to ensure the safety of the structure may rise strongly.*

REPLY: We thank the Reviewer for the precious suggestion and acknowledge the importance of defining an appropriate width for the IZ. Nevertheless, we would like to stress that this work only aims at introducing the IZ and its width as a first order geometric factor between an active and capable fault (ACF) and a linear infrastructure. The IZ concept strongly depends on the possible geometrical interference patterns between an ACF and a linear infrastructure (see our Figure 7). To provide an initial, indicative parametrisation of the ACF, we propose that fault scaling laws can be used. This helps assess the mappable area where the deformation effects associated with ACFs activity are expected to impact the infrastructure. We recognise that a reliable IZ characterisation would require more detailed investigations and techniques, such as those that we propose in Section 5.2 and in Table 2 of the manuscript. Such a thorough investigation would certainly minimize the risk of both overestimating the IZ and underestimating the actual fault dimensions (e.g., the width of its damage zone). Therefore, we consider that the parametrisation of the IZ proposed in this work could be considered as a first step to the first order assessment of the interference zone between the fault and the infrastructure.

*COMMENT 2: Regarding section 5.2 and discussion about remote sensing and identification of fault damage zones using microtopographic technics, I think it is important to precise that this approach only works if the topography signal of the fault surface rupture and damaging has been preserved in the landforms. I mean that if erosion of anthropic activities exceeds the expression of the fault-related deformation, this approach will not work, and that this limit deserves to be mentioned. Hence here comes the geophysical approach (nondestructive) and paleoseismological and geotechnical approaches (destructive).*

REPLY: We thank the Reviewer for this comment that we used in the revised manuscript to improve the description of the topographic techniques that can be used to identify the orientation and distribution of the fault pattern deformation. We agree that we should also highlight the limitations of these techniques. Lines 374-377.

*COMMENT 3: Another point is the organization of the sections in that chapter. Please make slightly more obvious the successions intro and scaling, RS, field geology and geophysics, then paleoseismology, dating and morphotectonics for slip rates assessments.*

REPLY: We thank the Reviewer for this advice. We have modified Section 5.2 to improve the flow and to better describe the sequence of techniques to be deployed to constrain the structural complexity of fault zones at a progressively increasing resolution.

**Reply to minor comments**

| COMMENT (row number and figure number refer to the early version) | REPLY (row number and figure number refer to the revised version without tracked changes) |
|---|---|
| L26, please specify the names of the Great Alaska and Niigita earthquakes | Ok, done. L 26. |
| L52 : Why not add water adduction structures ? e.g. Laquilla EQ, that has been strongly affected by a waterpipe rupture. | Thanks for the advice. Done. L 52. |
| L140: I suppose the processes here illustrated come from previous publications show them (e.g. Barnett 87, Peacock and Sanderson 91, 94). Would it be necessary to refer to the main ones in the caption ? | Thanks for the comment. Indeed, the schematic diagram of the two interacting and overstepping fault segments, as well as the graph in the upper part of Figure 4, are generally inspired by Figure 3 in *Peacock and Sanderson (1994)*. We fully agree with the Reviewer on the importance of including this reference within the caption, which has now been added to the manuscript. We have also cited *Kim et al. (2004)*, whose work we quote for the terminology used in the classification of damage zone types. L 142-143. |
| L155 : cumulative displacement is proportional to fault width with same order of magnitude ? They are proportional modulo a factor $10^2$ to $10^3$, what significance ? | Thanks for raising this point. We usually refer to Table 1 in *Scholz et al. (1997)*, which identifies 1:1 relationship between fault process zone dimensions and cumulative displacement based on a statistical correlation.
In our analysis, we compiled data from several published references with the aim of illustrating the spatial range of correlation between these two fault attributes. However, since we do not have access to the original datasets underlying the referenced studies, we cannot perform a new statistical regression and evaluate the significance. In this regard, we have also removed the reference line W=D from Figure 5b. |
| L156 : Why 2 orders io magnitude ? Mean 100 km fault will have damage zone 10 km ? There are no references to explain this scaling relation. | The reference to two orders of magnitude is based on the statistical regression documented by *Vermilye and Scholz (1998)*, which is also illustrated in Figure 5c of our manuscript. In their analysis, they reported that the width of the fault process zone typically scales with fault length by approximately two orders of magnitude. Specifically, other authors cited therein observed that faults c. 100 km in length are associated with damage zones approximately 1 km thick. L 158. |
| L181: Maximum displacement along a single fault trace ? Cause if one considers the fault system (here the two main fault segments and the damage zone within the overlapping zone. | Yes, in this sentence we are referring to the displacement profile observed along an individual fault segment within a fault system. L 180-181. |

| | |
|---|---|
| 189 : is it representative to consider a stand-alone fault segment ? Could you please bring proper example ? | Thanks for this thought. As we are talking about evolving ACFs, we believe that it could be important to also include the evolution of a single and isolated fault segment, especially in areas where active deformation is relatively young (see *Fossen and Rotevatn, 2016*). L 189. |
| L220: 14 % ? Do the author precise a range of length when this relation is observable ? This might depend on the seismogenic width (previously names fault height) ? There is a quite big discrepancy here ; in what extent could this discrepancy be addressed for definition of relay zones ? | Thanks for this insightful comment. We have added the range of fault lengths of *Acocella et al (200*0). In our study, we focus on the evolution and interaction of fault segments as mapped at the surface, without attempting to characterize the seismogenic source in depth. We acknowledge that the seismogenic width (formerly referred to as fault height) can significantly influence the scaling relationships between fault length, spacing, and interaction potential (e.g., *Walsh & Watterson, 1991; Peacock & Sanderson, 1991*). However, addressing this depth-related variability is beyond the scope of our work, which is restricted to the analysis of surface fault traces and their geometrical relationships, including relay zone development. L 222. |
| Figure 6 : the constant length model also corresponds to the characteristic slip model (Shwartz and coppersmith, 82). Is it necessary to precise it here ? | Thanks for the suggestion. While we recognize the conceptual overlap between the constant length model and the characteristic slip model as described by *Schwartz and Coppersmith (1984),* we chose not to explicitly refer to the characteristic earthquake/slip model in this context, as figure 6 does not aim to explore models of seismogenic source behavior. It focuses instead on the geometric and structural evolution of fault segments as observed at the surface. |
| Figure 7 : regarding the coseismic diffuse rupture, on the right we observe in the transfer zone any left step ridel (RL strike slip indication), instead on the left (within the tip zone), they are right step (left lateral strike slip indication). It might be more coherent to maintain the same pattern in the surface rupture scheme. | Ok, done. Figure 7. |
| L243: This sentence suggests you are prioritizing some parameters in the figure, but then it is not obvious that the different scenario presented in this figure allows you to choose parameters to prioritize. Maybe simply precise/reformulate the caption. | Thanks for pointing out this ambiguity. We have reformulated the caption. L 247-248. |
| Figure 8: Interesting. I found it hard to catch the location of the profiles displayed but I eventually understood | Thank you. |

| | |
|---|---|
| L249: certainly, but why ? | Thanks, we have added the justification. L 251-252. |
| L253: Unless I am mistaken, linear correlation in a log-log graph describe a power law correlation and not a linear one. | Yes, it is correct. Modified. L 256. |
| L255: "it is reasonable to expect", yes I suppose, but is there any references looking for cross-fault slip distribution (maybe from pixel correlation for instance) that may help to justify this assertion ? | Thanks for the suggestion. We have added a couple of references documenting coseismic slip profiles and deformation width during two earthquakes (*Barnhart et al., 2015* for the 2013 Mw 7.7 Balochistan earthquake and *Rockwell et al., 2022* for the 1999 Mw 7.1 Duzce earthquake). L 260. |
| L270: ''The length of the main slip surface'' is a bit confusing. I might suggest reformulating. | Thanks for the advice. It has been reformulated as "The length of the main fault surface". L 272-273. |
| L270 ''c.'' ? | Yes, we used it in the meaning of "about" or "around". L 273. |
| L272 : is there some relations describing the amount of displacement within the tip zones compared to the maximum surface slip observed ? This would help with assessment in the case of the near fault type scenario. Same question for the length of the tip growing ? | Thanks for the insightful question. Although the spatial distribution of displacement along faults – including the progressive decrease of slip toward fault tips – has been widely observed and qualitatively described (e.g., *Peacock & Sanderson, 1991; Nicol et al., 1996*), quantitative relationships specifically describing the proportion of slip within tip zones relative to the maximum surface displacement remain limited in the literature.

*Kim & Sanderson (2005)* provide a regression that links fault length to the area of tip damage zones (log(Tip damage area) = 1.0·log(fault length²) – 1.1), which indirectly reflects fault-tip growth processes, but does not directly quantify the displacement within the tip zones.

To our knowledge, no standard empirical relation currently exists that provides a fixed percentage of displacement within the first few meters of fault tips. Slip distribution is often modeled as elliptical or triangular, with tip values typically falling below 10–30% of the maximum slip (e.g., *Nicol et al., 1996*), but exact proportions vary with fault maturity, lithology, and local structural conditions.

We acknowledge that further empirical work is needed to better constrain these relationships and clarify their implications for surface faulting scenarios such as the near-fault case considered in our study. L 276. |

| | |
|---|---|
| L294: Yes, accounting for bloc rotation in relay zones is necessary. Here, quantitative information related to blocks' dimensions (and fractal aspect) would be useful. Also considering the rotation of tens of meter large blocks, this would generate at locale scale some surface rupture associated with offsets that could be analog to a classical surface displacement. Two processes can hence maybe be considered as analogs. | We thank the Reviewer for this comment. However, maybe we don't understand the meaning of the comment itself. Does the Reviewer refer to the combination of kinematic and dynamic ruptures during fault activation? Due to this misunderstanding, we preferred not to include this suggestion within the revised text. |
| L300 : I do not understand this criterion. Is it a concept of ''Bandes de réserve'', as buffer around the fault trace – a buffer that would be more pronounced on the hanging wall than on the footwall, according to previous observation on the relatively more damaged HW that FW ? In Morocco for instance, the regulatory Paraseismic Rules prescribe an arbitrary buffer zone 60 m from the fault trace and is associated to no implantation of new strategic and public-access buildings. Do you think any specific values might be appropriate for the carious scenario you defined here ? For instance, a relation between U, ang, damage zone width an length of the IZ ? | This concept is based on the Italian guidelines for seismic microzonation studies in areas affected by active and capable faults (*Technical Commission on Seismic Microzonation, 2015*). These guidelines define two main zoning elements:
• the Respect Zone: a fixed-width (30 m) no-construction buffer straddling the mapped fault trace, applicable only for faults with demonstrated surface rupture potential;
• the Susceptibility Zone: a broader zone (e.g., 160 - 300 m) that includes both the fault trace and surrounding areas potentially affected by primary and distributed faulting. This zone can be asymmetric, based on observed or inferred distribution of deformation, and varies in width depending on the certainty level and geometry of the fault.
The Susceptibility Zone is conceptually similar to the "Bandes de réserve", and asymmetry in this zone can be introduced when geological or geomorphological evidence indicates a preferential development of damage toward the hanging wall, as shown in Table A3 of the guidelines. L 313. |
| L311: I agree for structural architecture, but I would add that the surface rupture pattern also varies regarding to rupture type (e.g. supershear rupture will generate damages far away from main fault trace), rupture dynamics (propagation direction, directivity…) and local condition (water content, slope, ground cohesivity…). Those factors, combined, might reduce the ability to properly constrain the spatial distribution of coseismic displacement and extension of the damage zone. | Thanks for the advice. We have integrated its content within the text. L 327-328. |
| L322; This affirmation seems to be contradictory with explanations given in L300, L170 and L271. In fact, if the damage zone is | Thanks, rephrased. L 343-344. |

| | |
|---|---|
| asymmetric for thrust (with more damage in HW dur to strain distribution) but symmetric in a SS fault, I do not understand the meaning of this concluding sentence. | |
| L326: Those empirical laws correlate the amount of coseismic offset compared to magnitude or length and knowing the fault kinematics. They do not provide a direct empirical relation between the earthquake magnitude and the width of the core, and the fault damage zones (that as you explained before depending mainly on inheritance like local geology and structuration). For this, you must refer to the laws of figure 5. So, I would precise the sentence explaining that those laws can only give insight of the surface coseismic offset (mean and max), and that they depend of the fault kinematics. | We agree with this comment. We are referring to the coseismic displacement at the ground surface. We have modified the sentence. L 339-340. |
| L331: I follow the thinking. However, the mentioned laws bring uncertainty of three orders of magnitude (e.g. for a 1 m coseismic displacement, damage zone width might range from 0.1 m to 100 m. How to deal with such a large range to define | These relationships are intended to provide general, first approximation of fault zone dimensions, and are not sufficient on their own to define precise boundaries of the IZ. Therefore, they should be complemented by site-specific investigations to refine the fault model and, in turn, optimize the spatial definition of the IZ in applied contexts (such as the one we propose in Section 5 of the manuscript). L 335-336. |
| L334 : Color coded cell are often used for decision matrix. I suggest this as there are already green color on the 'high' | We thank the Reviewer for the suggestion. Done. Figure 10. |
| L347 : Geophysical survey could also be interesting, if combined with geological and geotechnical analysis. Also Geotechnical care could be useful to discuss damage zones and maybe identify surface rupture | Ok, added (here and later in the text). L 354 and 400. |
| L364: This section brings interesting technics to assess the surface rupture pattern and needs to be more referenced for each. | We thank the Reviewer for the suggestion. We added several references. L 373; 382; 386; 395; 402; 404-405. |
| L366 : To measure the height of fault scarp yes, but also horizontal displacement, cumulative or not. Also, it could be useful to precise the scales used for such surveys (dam, m, cm…). | We thank the Reviewer for the suggestion, added. L 382. See also L 374 and 380. |
| L371 : It seems that there are two confusions here, i. between photogrammetry (which uses stereographic correlation of optical images) and Lidar scanning, and ii. Between aerial and terrestrial. Both technics are useful, so this sentence might be simply reformulated. | We thank the Reviewer for the suggestion. The text was rephrased. L 386. |

| | |
|---|---|
| L375: please precise the parameters and bring examples (rises, crests, cross-correlation of depositional units in 3D paleoseismic trenches) | Done. L 390-391. |
| L377-78: This sentence is difficult to catch. What do you mean by slip surface(s), fault splays? Off-fault offset surfaces? Please reformulate and bring references. | We were referring to the (main) fault surfaces. L 395. |
| L378: Also please change section for geophysics, and precise that correlations with field data will be useful to conclude on the observed contrasts. | Added. L 398-400. |
| L384: please give references | This sentence has been removed in the revised version of the manuscript. |
| L395: You should mention OSL and radiocarbon dating and bring reference | Added. L 419. |
| L397: You should mention OSL and cosmonucleides dating (Be, Cl, He) and bring reference | Added. L 417. |
| L415-416: Please justify with existing examples or references that this interesting approach is properly compatible with the timing and budget requirement, as some dating, aerial acquisition, treatment and interpretation could be quite tight compared to delays usually running for infrastructure construction. | We thank the Reviewer for the advice. We prefer not to include references within the conclusion section. However, we have rephrased the sentence. L 440-443. |

---

## Author Response (AR2)

**RESPONSE TO THE REVIEWER'S COMMENTS**

**REV#2 (Dr. Robin Kurtz)**

**Reply to comments**

(row number and figure number refer to the revised version without tracked changes)

*COMMENT 1: Figure 6, still I think that the 'restriction' model (2) correspond to the characteristic slip, and hence Schwartz and coppersmith 82 might be mentionned. I do insist on this because the characteristic slip model is a very strong assuption used in deterministic (and indirectly in probabilistic) seismic hazard assessments.*

REPLY: We thank the Reviewer for the comment. We have modified the Figure 6 and added the reference in the text (Lines 200-201).

*COMMENT 2: L325. It would be appropriate to mention the notion of surface rupture (instead of or in complement to 'faulting style") and related patterns, actually sustained by structural setting.*

REPLY: We thank the Reviewer for the advice. Now we refer to "surface ruptures pattern" instead of faulting style (Line 327).

*COMMENT 3: L326. Typo issue, majuscule.*

REPLY: Ok, we have fixed it (Line 326).

*COMMENT 4: Fig 10: thanks for considering the colors. I meant as well to color 'low' and 'intermediate with adapted colors (e.g. yellow and red), and also to correlate the color 'temperature' with the level of danger, i.e. interraction with fault. Hence following this logic 'High' might be red and 'low' green. Well this is just a suggestion for display.*

REPLY: We thank the Reviewer for the suggestion. We have slightly modified the Figure 10 following the provided advice.

*COMMENT 5: L378, typo (maybe consider reading carefully the typo of this section, that has been widely developped).*

REPLY: We have slightly modified the sentence (Lines 379-381).